# Design of a New Vaccine Prototype against Porcine Circovirus Type 2 (PCV2), *M. hyopneumoniae* and *M. hyorhinis* Based on Multiple Antigens Microencapsulation with Sulfated Chitosan

**DOI:** 10.3390/vaccines12050550

**Published:** 2024-05-17

**Authors:** Darwuin Arrieta-Mendoza, Bruno Garces, Alejandro A. Hidalgo, Victor Neira, Galia Ramirez, Andrónico Neira-Carrillo, Sergio A. Bucarey

**Affiliations:** 1Doctoral Program in Forestry, Agricultural and Veterinary Sciences, South Campus, University of Chile, Av. Santa Rosa 11315, La Pintana, Santiago 8820808, Chile; darwuin.arrieta@uchile.cl; 2Escuela de Química y Farmacia, Facultad de Medicina, Universidad Andres Bello, 2320 Sazié, Santiago 8320000, Chile; b.garcsjarpa@uandresbello.edu (B.G.); alejandro.hidalgo@unab.cl (A.A.H.); 3Departamento de Medicina Preventiva, Facultad de Ciencias Veterinarias y Pecuarias, Universidad de Chile, Av. Santa Rosa 11735, La Pintana, Santiago 8320000, Chile; vneiraram@gmail.com (V.N.); galiaram@uchile.cl (G.R.); 4Laboratorio Polyforms, Departamento de Ciencias Biológicas, Facultad de Ciencias Veterinarias y Pecuarias, Universidad de Chile, Av. Santa Rosa 11735, La Pintana, Santiago 8320000, Chile; aneira@uchile.cl; 5Centro Biotecnológico Veterinario, Biovetec, Departamento de Ciencias Biológicas, Facultad de Ciencias Veterinarias y Pecuarias, Universidad de Chile, Av. Santa Rosa 11735, La Pintana, Santiago 8320000, Chile

**Keywords:** PRC, vaccine, microparticles, chitosan sulphated, PCV2, *Mycoplama hyopneumoniae*, *Mycoplama hyorhinis*, heparan sulfate receptors, mimetic

## Abstract

This work evaluated in vivo an experimental-multivalent-vaccine (EMV) based on three Porcine Respiratory Complex (PRC)-associated antigens: Porcine Circovirus Type 2 (PCV2), *M. hyopneumoniae* (Mhyop) and *M. hyorhinis* (Mhyor), microencapsulated with sulfated chitosan (M- ChS + PRC-antigens), postulating chitosan sulphate (ChS) as a mimetic of the heparan sulfate receptor used by these pathogens for cell invasion. The EMV was evaluated physicochemically by SEM (Scanning-Electron-Microscopy), EDS (Energy-Dispersive-Spectroscopy), Pdi (Polydispersity-Index) and zeta potential. Twenty weaned pigs, distributed in four groups, were evaluated for 12 weeks. The groups 1 through 4 were as follows: 1-EMV intramuscular-route (IM), 2-EMV oral-nasal-route (O/N), 3-Placebo O/N (M-ChS without antigens), 4-Commercial-vaccine PCV2-Mhyop. qPCR was used to evaluate viral/bacterial load from serum, nasal and bronchial swab and from inguinal lymphoid samples. Specific humoral immunity was evaluated by ELISA. M-ChS + PRC-antigens measured between 1.3–10 μm and presented low Pdi and negative zeta potential, probably due to S (4.26%). Importantly, the 1-EMV protected 90% of challenged animals against PCV2 and Mhyop and 100% against Mhyor. A significant increase in antibody was observed for Mhyor (1-EMV and 2-EMV) and Mhyop (2-EMV), compared with 4-Commercial-vaccine. No difference in antibody levels between 1-EMV and 4-Commercial-vaccine for PCV2-Mhyop was observed. Conclusion: The results demonstrated the effectiveness of the first EMV with M-ChS + PRC-antigens in pigs, which were challenged with Mhyor, PCV2 and Mhyop, evidencing high protection for Mhyor, which has no commercial vaccine available.

## 1. Introduction

The pork industry is mainly affected by pathogenic microorganisms, which frequently include those that cause the Porcine Respiratory Complex (PRC) [1,2]. The PRC is a dynamic process of a multifactorial disease; it causes serious losses due to damage to the tissues of the respiratory system of animals, affecting their physiology, generating high morbidity and mortality at various stages of the productive cycle. In pigs, PRC mainly affects the rearing and fattening phases [1,2].

It has been described that the interactions between environmental factors [1,2,3], factors corresponding to the host, such as its immune status (among others), and factors associated with the virulence of the primary and secondary infectious agents are the main factors for the onset of PRC. Infective agents such as Porcine Circovirus Type 2 (PCV2), *Mycoplasma hyopneumoniae*, and Porcine Reproductive and Respiratory Syndrome Virus (PRRSV) are those that initially settle in the pig’s respiratory system, generating the infection [1,3,4]. These pathogens predispose to a co-infection with secondary pathogens such as *Mycoplasma hyorhinis* (among other microorganisms), which is found in the bacterial microbiota of the upper respiratory epithelium and which take advantage of the imbalance to proliferate [1,5,6].

PCV2 settles in the pig´s respiratory system, causing the first lesions, generating infection, and facilitating the coinfection with secondary pathogens. Secondary infections are usually produced by opportunistic bacterial microbiota of the respiratory epithelium. PCV2 infection may downregulate the host immune system, paving the way for infection by other pathogens and increasing the severity of disease in pigs [7].

*M. hyopneumoniae*, the primary etiological agent causing chronic swine enzootic pneumonia, is the first to colonize the respiratory tract producing adherence to hair cells of the trachea, bronchi and bronchioles, generating loss of cilia and consequently massive colonization of the respiratory system [8,9].

*M. hyorhinis* is responsible for polyserositis, pneumonia, and arthritis in piglets. Recently, *M. hyorhinis* has been recognized as a ubiquitous but relevant pathogen of the upper respiratory tract in recently weaned pigs. In addition, *M. hyorhinis* is concomitant with other infections, such as those produced by *M. hyopneumoniae* and PCV2 [5,6]. However, its exact role in the etiology of PRC has not yet been established [10].

Vaccination is essential to control PRC, and various options are on the market, especially to combat pathologies caused by PCV2 and *M. hyopneumoniae* [6,11]. For *M. hyopneumoniae*, the commercial vaccines are widely used with discrete cost-efficient ratios. Available vaccines induce only partial protection and do not prevent infection [12]. Currently in Chile, there are 22 registered vaccines against pathogens associated with PRC; however, these commercial vaccine options do not include a protective response against *M. hyorhinis* [13]. *M. hyorhinis* is an active agent producing PRC, generating specific lesions in piglets and, as a consequence, great economic losses for the farms [14,15]. There are few reports of vaccine development against *M. hyorhinis*, and it is still in the experimental phase, with no vaccines commercially available [10,16,17,18]. It should also be noted that the governmental authorities in Chile are implementing a comprehensive control and eradication plan for PRRSV [19].

The use of vaccine adjuvants has improved the immunization of pigs [12,20]. This is the case for chitosan, obtained by partial deacetylation of chitin found in crustacean shells and insect cuticles [21]. Chitosan has been described as a vaccine adjuvant and as a vehicle for antigen encapsulation. Chitosan protects antigens from enzymatic degradation while releasing antigens in a controlled manner and promoting their absorption [21,22,23,24]. Other properties of chitosan include its biocompatibility, mucoadhesion, promotion of absorption, antimicrobial activity [22,25,26] immunomodulating capacity [27,28] and reduction of cargoes toxicity [21,29].

The use of chitosan polymers, in suspension or in microparticle systems (MPs), increases the encapsulation efficiency of antigens and regulates their release from the MPs [27,30,31,32]. Among the characteristics that determine the functional properties of chitosan are the degree of deacetylation (percentage of amino groups that remain free in the chitosan molecule) and the average molecular weight [22,33,34]. pH also affects the mucoadhesive properties of chitosan, as this polymer is positively charged at acidic pH due to its pKa of 6.5 [22,33,35]. 

The functionalization of chitosan has been proposed. This is the case of sulfated chitosan (ChS), which promotes the mucoadhesive properties of chitosan, also improves its permeability and stability and allows greater control over the release of microencapsulated macromolecules [36]. In addition, ChS mimics the cellular heparan sulfate (HS) receptor. Therefore, ChS is proposed as a decoy, a blocker and a vehicle for various pathogens that use the HS receptor as a mechanism of cellular attachment and invasion. Pathogens that use HS as a receptor include primary and secondary microorganisms associated with PRC, such as PCV2, PRRSV, *M. hyopneumoniae and M. hyorhinis* [36,37]. The objective of this research was to evaluate the effectiveness of an experimental trivalent vaccine against PCV2, *M. hyopneumoniae* and *M. hyorhinis*, based on microencapsulation of antigens with sulfated chitosan, and to provide preliminary evidence on its contribution to the prevention of PRC.

## 2. Materials and Methods

### 2.1. Vaccine Microparticles

The vaccine microparticles (MPs) were formulated by spray-dryer atomization technique by using a Buchi Mini Spray Dryer B-290. The antigens sources included formalin-inactivated *M. hyopneumoniae* (strain ATCC25934TM) and *M. hyorhinis* (strain ATCC 17981TM) as well as yeast-produced PCV2 virus-like particles (VLP) (*S. cerevisiae*, strain N30/pYES2:cap) as described by Bucarey et al. [23]. The solution used for formulation of MPs contained 1% p/v of sulfated chitosan and 0.5 mg/mL of each antigen. 

The resulting microparticulate formulation was collected and weighed. The efficiency of the microencapsulation process was 90–95% (estimated by subtracting the total amount of antigens remaining in the formulation from the initial amount of antigen added to sulfated chitosan solution). Antigens concentrations were measured using the BCA™ protein assay kit (Pierce, Rockford, IL, USA). Samples of microparticles were suspended in PBS (pH 7.0; final concentration, 1 mg/mL) and stored at 5 °C until animal inoculation.

### 2.2. Morfoestructural by Scanning Electron Microscopy (SEM) and Elemental Analysis by Energy Dispersive Spectroscopy (EDS) of Vaccine Prototypes

Morphostructural and elemental analysis was carried out using a Microscope (JEOL instrument model JSM-IT300LV, Pleasanton, CA, USA) coupled to an X-ray energy dispersive detector located at the Faculty of Dentistry of the University of Chile. PRC-antigens (*M. hyorhinis, M. hyopneumoniae* and PCV2) were added to the MPs samples of Chc (commercial chitosan) and ChS. Before evaluation, they were metallized with gold, according to methods previously described [23] to examine the morphology and size of the individual microparticles. 

### 2.3. Analysis of Surface Charge (Zeta Potential) and Polydispersity Index (PdI) of Microparticles of Vaccine Prototypes

The MPs loaded with antigens were characterized in a particle size and zeta potential analyzer (model Zetasizer nano Zs, Malvern, Spain). The determinations of the different MPs were carried out at a concentration of 0.5 mg/mL, under magnetic stirring, followed by sonication, during 30 and 15 min respectively. Additionally, the Chc + PRC-antigens and ChS + PRC-antigens solutions were centrifuged at 5000 rpm for 5 min (Hemrle, model Z36HK).

### 2.4. Experimental Design

The tests were carried out in animals under controlled environmental conditions in the Animal Management Unit of the Faculty of Veterinary and Livestock Sciences (UMA-FAVET), of the University of Chile. The animals were located in spaces with total dimensions of 40 m^2^ (5 m × 8 m), whose pens had dimensions of 2.2 m^2^ (1 m × 2.2 m). Each subdivision had its own entrance, feeder and drinking fountain, according to the number of animals that were housed in the facility. All facilities have been properly sanitized and disinfected before, during and after the animals’ stay.

Twenty male pigs (*Sus scrofa*) of race Landrace were used including three-week-old pigs (post-weaning) from conventional farms free of porcine reproductive and respiratory syndrome virus (PRRSV) and swine influenza virus (SIV). This pig herd had a history of animals with clinical signs of PRC. The parental herd, from which the pigs used in the experiment came, received the following vaccines: PCV2 subunit vaccine and inactivated *M. hyopneumoniae* (CIRCUMVENT^®^ PCV M, MSD, Merck & Co., Inc., Rahway, NJ, USA); *Actinobacillus pleuropneumoniae* bacterin-toxoid vaccine (PORCILIS^®^ APP, MSD, Merck & Co., Inc., Rahway, NJ, USA); inactivated vaccine *Erysipelas*, Porcine Parvovirus, *Leptospira canicola*, *L. icterohaemorrhagiae*, *L. australis*, *L. grippotyphosa*, *L. Pomona* and *L. tarassovi* (PORCILIS^®^ ERY + PARVO + LEPTO, MSD); inactivated *Lawsonia intracellularis* vaccine (PORCILIS^®^ ILEITIS, MSD); Live Porcine Rotavirus vaccine, inactivated *Escherichia coli* and *Clostridium perfringens* type C toxoid (PORCILIS^®^ RCE, MSD). The immunization protocol and frequency of administration of these vaccines in the pig farm were carried out according to the manufacturer’s recommendations. 

The animals were moved to the UMA-Favet and distributed randomly into 4 groups of 5 individuals each and identified by application of color in the dorsal region. Each experimental group of animals (4 in total) was located in a subdivision, that is, 4 subdivisions with 5 pigs each. The animals were evaluated through a period of 12 weeks. The 4 groups of pigs were treated as follows: Group 1 received the experimental formulation that consisted of a microparticle lyophilized (MPs) of ChS, loaded with aprox. 50 ug/dose of each antigen (PCV2, *M. Hyopneumoniae* and *M. Hyorhinis*), resuspended in PBS to 1 mg/mL final concentration of MPs. A 1 mL dose (1 mg/mL) was administered to animals intramuscularly. Group 2 received the same experimental formulation as group 1, administered orally (2 mL dose of 1 mg/mL formula) and nasally (1 mL dose of 1 mg/mL formula) (O/N). Group 3 received the placebo treatment consisting of the experimental formulation without pathogenic antigens (MPs of sulfated chitosan resuspended in PBS, without antigens), final concentration MPs 1 mg/mL. Group 4 received a commercial vaccine treatment, equivalent to the commercial vaccine Suvaxyn^®^ (Zoetis, Santiago, Chile) Gold PCV MH (SAG registration number 2494-B). No clinical symptoms and local reactions at the site of vaccine administration were observed. 

The treatments used are shown in Table 1.

### 2.5. Controlled Conditions for Animals

Pigs were acclimatized for 10 days in spaces with controlled conditions (UMA-FAVET), physically examined and blood-tested. In addition, nasal swab were taken on arrival, to test for pathogens to which they were challenged (Table 2). All animals were in good clinical conditions as they were free of external lesions, respiratory conditions, gastrointestinal signs and presented normal temperature. The body condition and body temperature were measured once a week, while the presence of lesions, gastrointestinal and respiratory symptoms were evaluated daily. The pigs received food and water ad libitum during the trial. The feeders and drinkers were inspected daily until the end of experimental period (12 weeks). The exhaustive cleaning of facilities was carried out 3 times a week. Individuals whose treatment corresponded to intramuscular immunization (IM) were vaccinated with 1 mL by injection into the muscles of the lateral region of the neck. Orally and nasally (O/N) immunized individuals were vaccinated with a total volume of 3 mL. Using a syringe without needle, 1 mL was directly applied to each nostril and 1 mL directly applied to the oral cavity. In addition, all experimental groups received two boosters of their respective treatments 21- and 42-days post immunization (DPI), respectively (see Table 2).

### 2.6. Experimental Challenge

In week 7, after starting the experimental treatments (T1), the challenge was carried out by inoculating the four study groups (Table 2) with virulent strains of *M. hyopneumoniae, M. hyorhinis* and PCV2. The inoculum was obtained from lung maceration of pigs with clinical signs and histopathological lesions compatible with PRC, from a commercial herd (the same commercial herd where the pigs used in the experiment came from). For this, the injured tissues of the middle lung lobe were macerated, diluting them in 50 mL of 1X PBS without antibiotics. The macerate was centrifuged in 50 mL conical tubes and the supernatant was used as inoculum. Subsequently, an aliquot of the macerate was extracted for DNA extraction and then the multiplex real-time PCR was performed for PCV2, *M. hyopneumoniae* and *M. hyorhinis*, using the primers and probes described in Table 3, with which the presence of the three pathogens is confirmed. The estimated levels of each pathogen in the inoculum were calculated for *M. hyopneumoniae*, *M. hyorhinis* and PCV2 and was set at a value of Ct < 15. The three lungs with the lowest Ct for *M. hyopneumoniae, M. hyorhinis* and PCV2 were selected for further analysis. Additionally, the inoculum macerate was checked by PCR negative for porcine reproductive and respiratory syndrome virus (PRRSV), swine influenza virus (SIV), Porcine Parvovirus and *Actinobacillus pleuropneumoniae*. 

### 2.7. Intratracheal Inoculation

The sedation of animals was performed with ketamine (8 mg/kg) and xylazine (4 m/kg), intramuscularly injected in the lateral region of the neck. The sedative effects were observed between 3 and 6 min after injection; during this time period, the animals were raised, held vertically, and inoculated. To inoculate each animal, a mouth opener was used in the snout of the animals, allowing the visualization of the larynx with a Miller type laryngoscope using a sheet (N° 4). Later, a 18FF aspirate probe for intratracheal inoculation was used. With a 10 mL syringe, 10 mL of inoculum were released through the probe, followed by 5 mL of PBS 1X without antibiotics. After inoculation, the animals were observed until complete recovery from sedative effects. No clinical symptoms or signs were observed after inoculation and throughout the course of the experiment. 

### 2.8. Obtaining and Processing of Samples

Seven samples were obtained, each from a different time from T0 though T6 (Table 2). They consisted of individual samples of nasal swabs (sample times T0 to T5) or bronchial samples (sample time T6, *postmortem*), sample of inguinal lymph node (sample time T6, *postmortem*) and blood (sample times T0 to T6). The blood samples were obtained from the jugular vein, using a 23 G caliber needle connected to a vacutainer. The samples were processed immediately after being obtained and frozen at −20 °C until use. For nasal swabs, sterile nasopharyngeal torulas were introduced into both nostrils of the nose and stored in 5 mL pipes with viral maintenance medium. These samples were maintained at 4 °C before storing at −20 °C for later processing.

Euthanasia and necropsy were performed at the end of the experiment (Experimental week N° 12). The experiment was authorized by the Institutional Committee for the Care and Use of Animals, University of Chile. Animals were euthanized by intravenous overdose of pentobarbital. The AVMA (American Veterinary Medical Association) Euthanasia Guide suggests the use of 1 mL/5 kg of body weight. The samples obtained during the necropsy (T6) were individual samples of inguinal lymph nodes, bronchial swabs and blood.

### 2.9. Detection of the Presence of PCV2, M. hyopneumoniae and M. hyorhinis by PCR (Real Time)

Extraction and purification of the DNA was performed using two extraction kits. The GeneJET Genomic DNA kit (Thermo Scientific™, Waltham, MA, USA) was used for the first 3 samples, and the PureLink™ Genomic DNA Mini Kit and Extraction Kit (Invitrogen, Waltham, MA, USA) for the last four sampling times. Three sets of primers were used for detection of the pathogens (one per pathogen). The primers were used to detect PCV2 by amplifying the gene encoding for replicasa (REP), located in the open reading frame 1 (ORF1). Primers to detect *M. hyopneumoniae* are modified versions of those described by Marois et al., 2010 [38] to amplify the p102 gene. Detection of *M. hyorhinis* was carried out by amplifying the 16S coding gene, as described by Stakenborg et al., 2006 [39]. A positive control DNA extracted from the bacteria/virus was used. Detection of PCV2 and *M. hyopneumoniae* via qPCR Multiplex and for the detection of *M. hyorhinis*, Sybr Green was used (Table 4). In addition, a melting curve from 60 °C to 95 °C, with increases of 0.5 °C, was used. After the amplification was performed, its positivity was confirmed by showing the amplified product.

### 2.10. Evaluation of the Induction of Antibodies for PCV2, M. hyopneumoniae and M. hyorhinis through ELISA Test 

Blood samples from pigs were centrifuged at 4500 rpm for 5 min, and the serum was collected and stored at −20 °C until use. Antibodies against PCV2 were identified using an antibody test kit (Biocheck, South San Francisco, CA, USA) according to the manufacturer’s instructions. For *Mycoplasma hyopneumoniae*, the ID Screen^®^
*Mycoplasma hyopneumoniae* competition (ID.VET™, Grabels, France) was used according to the manufacturer’s instructions. For *M. hyorhinis*, we used a homemade indirect ELISA test (since no detection kits are available for this pathogen). To perform the ELISA, a Nunc MaxiSorb plate (Invitrogen™) was first sensitized with a 40 µg/mL formalin-inactivated *M. hyorhinis* solved in carbonate buffer (5 mM; pH 9.4), with 100 µL added to each well. In addition, controls were added in triplicate for the primary antibody (1:1000 dilution of serum sample positive for *M. hyorhinis* antibodies), secondary antibody (1:2500 dilution of goat anti-swine IgG-HRP conjugate; KPL, St. Kalamazoo, MI, USA) and without antigen (carbonate buffer), respectively. Once the plate was prepared, it was incubated for two hours at 37 °C in an oven. Three washes with 100 µL PBS-Tween 0.05% were performed between each step.

After the first incubation, the plate was blocked with PBS-T (PBS-TWEEN 0.05%) with 3% BSA (Sigma-Aldrich Co., Ltd., St. Louis, MO, USA) for two hours at 37 °C. After blocking, primary antibody incubation was performed using a 1:150 dilution of the serum samples in PBS-T, and 100 µL was added to each well (samples were analyzed in duplicate). Controls were incubated with PBS-T, and the plates were incubated for 2 h at 37 °C or overnight at 4 °C. 

For secondary antibodies, 100 µL of goat anti-swine IgG-HRP conjugate (KPL) was added to each well at a 1:2500 dilution. Controls were incubated with PBS-T at 37 °C for one hour. To reveal the plaque, 50 µL of the 1-Step™ Ultra TMB solution (Thermo Scientific™) was added to each well and allowed to incubate at 37 °C until a color change was observed or a maximum of 20 min. To stop the reaction, 50 µL of stop solution (1 M or 2 N H_2_SO_4_) was added. Finally, the plate was read at 450 nm within the next 30 min.

### 2.11. Statistical Analysis

Results obtained by indirect ELISA analysis were entered, sorted, graphed and statistically analyzed using Prism-GraphPad™ (Version 10, GRAPHPAD SOFTWARE). The normal distribution of the variables in each group and treatment was evaluated using the Shapiro–Wilk test. For the analysis of variances with normal distribution, *p*-value was determined with significance level of 0.05; the repeated measures ANOVA test (RM-ANOVA) was used together with Tukey’s correlation to compare and adjust for antibody titer variables between groups. For those results that did not present a normal distribution, that is, *p*-value ≤ at the 0.05 significance level, the Friedman test was used along with the Dunn test to correct for multiple comparisons.

## 3. Results

### 3.1. Morfoestructural by Scanning Electron Microscopy (SEM) and Elemental Analysis by Energy Dispersive Spectroscopy (EDS) of Vaccine Prototypes

The SEM results demonstrate that vaccines formulated with Chc + PRC-antigens, which are characterized by presenting spherical microparticles of 1.2–6.5 µm with an irregular surface with concavities and grooves, as shown in Figure 1A. The vaccines formulated with ChS + PRC-antigens are mainly spherical and to a lesser extent spheroidal in morphology; they have a smooth surface with concavities of sizes from 1.3 to 10 µm (Figure 1B). As shown in Figure 1B, the coexistence of smooth spherical particles with simple and multiple concavities is verified. These characteristics were consistently observed in several observations (*n* = 3) only with the ChS formulations.

#### 3.1.1. Elemental Analysis by Energy Dispersive Spectroscopy (EDS)

EDS allows verifying that in the samples ChS + PRC-antigens, C and O are the main components, N has a component less than 10% and S has a concentration less than 5% (Table 5). As a control, the results obtained with EDS of MPs formulated with Chc + PRC-antigens are presented in Table 5, where it is observed that C is the main element (68%), followed by O (25%) and then N (6.81%), without the apparent presence of S atoms. These results show the greater presence of S atoms in ChS, compared to the Chc control (non-sulfated).

#### 3.1.2. Analysis of Surface Charge and Polydispersity of Microparticles of Vaccine Prototypes

The results of the determination of size and zeta potential (ζ) of the microparticles of the prototype vaccine show that the surface charge patterns, measured by zeta potential, remained constant and yielded markedly negative values (−33.2 mV on average) in all populations of microparticles formulated from ChS (Table 6), which was consistent with the chemical functionalization of chitosan by sulfate groups. At the same time, it was observed that they presented a low polydespersity or variation (0.4 average).

### 3.2. Experimental Challenge of Animals under Controlled Conditions

According with the experimental design, animals were challenged with three pathogens associated with PRC (PCV2, *M. hyopneumoniae* and *M. hyorhinis*). Therefore, DNA detection of PCV2, *M. hyopneumoniae* and *M. hyorhinis* by PCR-Real Time was performed before starting the experimental treatments. The detection of these pathogens involved in this study (see Table 2) (time T0) was carried out with the objective to know the health status of the animals at the time of arrival at the experimental units in a controlled environment by evaluating the existence of natural carriers of PCV2, *M. hyopneumoniae* and *M. hyorhinis* during the experimental procedure treatments (T1~T5).

The results indicate that PCV2 and *M. hyopneumoniae* were not present in experimental animals. However, given that these pigs come from a commercial farm, whose parental herd presented clinical signs and histopathological lesions compatible with PRC, it is expected that pigs present antibodies against these pathogens, probably due to previous contact with these microorganisms. The case of *M. hyorhinis* was diametrically different, since the ubiquitous presence of this pathogen was observed in T0 and T3 samples, which was reduced in later times, indicating lower recurrences of infection. It is important to mention that no clinical symptoms associated with PRC were observed after inoculation and throughout the course of the experiment.

Finally, sampling time T6, which correspond to necropsy of the animals, allowed testing for bronchial swab and inguinal lymph node samples. These samples are essential to determine infectious processes occurring in the deep respiratory tract by pathogens such as *M. hyopneumoniae* and *M. hyorhinis*, or lymph nodes for PCV2. These data are represented in Table 7.

The bronchial swab simples have a greater diagnostic value for *M. hyorhinis* than for PCV2 and *M. hyopneumoniae* because *M. hyorhinis* is a ubiquitous commensal pathogen in the upper respiratory tract and can only colonize the lower respiratory track after damaged is produced, particularly in the mucociliary apparatus, thereby facilitating its spread. Samples of inguinal lymph nodes are valuable indicators of PCV2 infection because of tropisms of PCV2 for lymphoid tissue, where it actively replicates after infection of respiratory tissue, moreover, affecting the immune system of the host.

As shown in Table 8, detection of pathogens for the controls (groups 3 and 4) had a similar behavior against PCV2 and *M. hyorhinis*, but not for *M. hyopneumoniae*, where the commercial vaccine (group 4) and the the experimental formulation (Groups 1 and 2) were found to be more effective. The experimental formulation showed administration route effects over PCV2. The experimental formulation was more efficient when administrered intramuscularly (IM), obtaining the best result against PCV2 (group 1), compared with the other groups.

The oral/nasal (O/N) route of administration was the less effective among the study groups. The case of *M. hyopneumoniae* infection was different, as the results were promising, regardless of the route of administration (see Table 8). However, the best results were obtained by O/N route, with an equal value to that of the commercial vaccine (group 4), without individuals positive for *M. hyopneumoniae*. While in the intramuscular administration only one positive animal was observed, which was still better than the placebo control.

The effectiveness of the experimental formulation against *M. hyorhinis* administered through O/N route (group 2) is lower (Table 8), presenting three positive individuals at the end of the in vivo test, compared with the controls (Groups 3 and 4). Importantly, the intramuscular formulation was the only one that did not present positive animals for *M. hyorhinis*.

#### Evaluation of Induction of Specific Antibodies against PCV2 and *M. hyopneumoniae* and *M. hyorhinis* by ELISA Test

Induction of antibodies targeting each pathogen was evaluated using different ELISA techniques due to kits availablility for each pathogen. It is worth mentioning that no analyses were carried out for sample times T5 and T6 because the experimental challenge was carried out one week before sample time T4, where the results of the ELISA tests might be affected by pathogen inoculation in this study. These results are summarized in Figure 2.

All experimental groups had a high titer of antibodies against PCV2 at the beginning of the study. This may be due to the immunity that the piglets obtained during the suckling period (considering that the parental herd was vaccinated against PCV2), but the titers decreased rapidly over time. This effect is clearly evidenced in group 3 (Figure 2C), which, without a stimulus, presented an abrupt reduction in antibodies, without a subsequent increase.

The reduction in antibodies is attenuated by the commercial vaccine and experimental formulation (IM and O/N) (Figure 2A,B,D) from T2, whose titers remain constant in T4, but increases with the experimental challenge. This effect was less evident in the O/N route (Figure 2B). However, it was observed that the elevation of antibody titers was more homogeneous with the administration of the experimental formula by IM route, compared to the experimental formulation route O/N and the commercial vaccine.

To evaluate the generation of antibodies against *M. hyopneumoniae*, a diagnostic test was used qualitatively. Because this ELISA test is of a competitive type, the signal/noise ratio is inversely proportional to the amount of antibodies in the sample. The results are summarized in Figure 3.

Figure 3 describes that the antibody titers for *M. hyopneumoniae* in the controls (group 3 and 4) increased, but was only significant in the commercial vaccine at T2, then decreased abruptly and then increased between T3 and T4 probably due to the experimental challenge. The experimental formulations also significantly increased antibody titers at T2, but their decrease was less abrupt at T3 compared to pigs receiving the commercial vaccine (Figure 3A,B,D). Greater homogeneity was observed in the increase of antibodies in group 2 (O/N route), compared to group 1 (IM route) and group 4 (commercial vaccine).

The results for evaluation of the generation of antibodies against *M. hyorhinis*. are presented in Figure 4.

Antibody titers for *M. hyorhinis* were similar in groups 3 and 4 (controls), probably due to what was mentioned above, since it is a commensal bacterium in pigs, who are constantly exposed to this pathogen, constantly activating the immune system against this pathogen, subsequently generating immunotolerance or a poor immune response. However, in the animals that received the experimental formula in both routes of administration (IM and O/N), a significant increase in antibody titers against *M. hyorhinis* was observed (Figure 4A,B), which suggests that the experimental formulations induced greater immunogenicity for this pathogen compared with the commercial vaccine.

## 4. Discussion

The results of SEM, EDS, PdI and zeta potential contributed to evaluating the prototype vaccine in the present investigation. In general, the behavior of chitosan MPs (both in vitro and in vivo) depends on physicochemical properties such as size, zeta potential and surface area characteristics [40,41,42,43].

The SEM showed that the formulation with MPs of ChS + PRC-antigens had a size between 1.3 and 10 μm (Figure 1B) and showed a low Pdi (Table 6), which is an important factor in immunization because efficient uptake by M cells requires microparticles measuring <10 μm in diameter to reach the dome of Peyer’s patches [44], as previously discussed by Bucarey et al. [23]. Other authors reported results of studies with biodegradable artificial antigen-presenting cells showing that 8 μm microparticles led to significantly greater activation of antigen-specific CD8+ T cells than 130 nm nanoparticles [45], which is important when it is necessary to obtain an immune response against strict or facultative intracellular pathogens, such as those of the present study.

In recent studies, Canelli et al. [46] tested on pigs a dry powder nasal vaccine, obtained from the deposition on a solid carrier of an inactivated antigen (*M. hyopneumoniae*) and a chitosan-coated nanoemulsion as an adjuvant. In their results they reported that the surface charge of these nanoemulsions, measured as zeta potential, ranged from 9 to 21 mV and was attributed to the presence of chitosan. Those are also similar to the results of the present work, where the control group of Chc + PRC-antigens presented a positive surface charge (Table 6), which coincides with results from previous experiments in our laboratory using microencapsulated PCV2 antigens with low molecular weight chitosan. In that case, surface charge was also positive [23], which suggested that a positive zeta potential is beneficial for uptake and transport in M cells because the membrane of M cells is negatively charged [47]. However, it contrasts with the zeta potential values of the MPs of ChS + PRC-antigens, whose surface charge (Table 6) was markedly negative (−33.2 mV on average) as a consequence of the chemical functionalization of chitosan by sulfate groups, evidenced when analyzing the EDS results (see Table 5).

It has been described that the degree of sulfation and the presence of anionic groups (among other variables) of sulfated polysaccharides contribute to their antiviral activity [48]. Other studies indicate that hypersulphated polysaccharides interfere with electrostatic interactions between the positively charged region of the viral envelope glycoproteins and the negative charges of the HS surface receptor chains [37,49], which is important to consider in the present experiment in relation to PCV2 because the surface charge of the experimental formula was markedly negative (Table 6). We would suggest that the probability of electrostatic attraction or interference between the positively charged regions of the PCV2 glycoprotein (with which the animals were challenged), decreased probability of viral binding to the HS cellular receptor. The hypothesis suggested above could also be transferred to the rest of the pathogens (*M. hyopneumoniae* and *M. hyorhinis*) used in this trial because they use the HS cellular receptor for intracellular entry [36], considering the non-detection obtained (negative PCR-Real Time) with the experimental formulation (Table 7 and Table 8) for non-viral pathogens (*M. hyopneumoniae* and *M. hyorhinis*) in the vaccinated and challenged animals of the present study.

It should be mentioned that despite its negative surface charge, the functionalization of chitosan has been proposed and the hypothesis suggests that ChS promotes the mucoadhesive properties of chitosan, also improves its permeability and stability and allows greater control over the release of microencapsulated macromolecules [36]. Those properties could contribute to improving the interaction between immunocompetent cells and microencapsulated antigens, such as those administered in the experimental formulation of the present study. However, at present there is no clear consensus about molecular weight of chitosan on other biological effects such as antimicrobial properties, with even some contradictory observations [26,37,50,51].

Nevertheless, it should be noted that the information that currently exists on these parameters in this vaccine prototype is scarce [23,46] to properly discuss or compare with results from other research on experimental vaccines, in part because this study is the first with a trivalent vaccine prototype (*M. hyorhinis*, PCV2 and *M. hyopneumoniae*) using antigens microencapsulated with sulfated chitosan.

This study also considered the effect of the administration (intramuscular vs. oral/nasal) in the response to antigens, considering immunization, challenging and evaluating in a corresponding period to the transition into rearing cycle. It is at this stage when PRC is described as a frequent clinical presentation that negatively impacts the health of pig production systems [1,3,7].

The reduction in maternal antibodies during the experimental time course makes it an appropriate timing to evaluate the vaccine prototypes. Preclinical evaluation of animals regarding viral and bacterial load was carried out before receiving the vaccine to finally evaluate humoral immune responses in challenged animals under controlled conditions. This study was based on the information collected in the formulation described by Bucarey et al. [36] for a vaccine against PCV2 which has similar characteristics to the experimental formulation evaluated with the exception that the vaccine evaluated in this work incorporated bacterin-like antigens for *M. hyopneumoniae* and *M. hyorhinis*. Previous results indicated that vaccine formulations could contain PCV2 and *M. hyopneumoniae* antigens separately, but not as a multivalent vaccines [36].

Another difference with the previous study [36] is that the prototype vaccine was not intended as a parenteral vaccine, emphasizing the fact that the vaccine has greater advantages when administered O/N due to its mucoadhesive properties of ChS. However, the low stability of antigens exposed to the harsh conditions in the gastrointestinal tract, together with the induction of mucosal tolerance, make difficult the induction of a reliable immune response through oral/nasal delivery. Thus, two boosters were required in this study at 21 and 42 days after the first immunization, the second one 7 days after the experimental challenge

Other studies [52,53,54,55,56], describe multivalent vaccines containing antigens against *M. hyopneumoniae* and PCV2. In addition, other authors described the possible use of multivalent vaccine antigens PCV2 and *M. hyopneumoniae* [57,58,59], while other researchers propose further combinations with additional antigens that come from a microorganism that causes disease in pigs, including *M. hyorhinis* antigen [10].

An inactivated bivalent vaccine against *M. hyorhinis* and *M. hyopneumoniae* has been previously assessed for safety, immunogenicity, and efficacy against *M. hyorhinis* infection. No changes were reportedly measured in clinical signs or rectal temperature after vaccination. Significant differences in macroscopic lung lesion scores between vaccinated and unvaccinated pigs after challenge demonstrate the efficacy of vaccinating with inactivated *M. hyorhinis*. Interestingly, high antibody response against *M. hyorhinis* and *M. hyopneumoniae* was only observed in vaccinated pigs [16], which coincides with the titers of antibodies against *M. hyorhinis* and *M. hyopneumoniae* obtained with the administration of the experimental vaccine in the present trial (Figure 3 and Figure 4).

Other trials [10] evaluated a combined PCV2 and *M. hyorhinis* inactivated vaccine in a model of PCV2 and *M. hyorhinis* infections. *M. hyorhinis* was isolated from the lungs of all unvaccinated pigs after challenge with *M. hyorhinis*; however, *M. hyorhinis* was not present in pigs from the immunized group after challenge with *M. hyorhinis*. In addition, the vaccinated pigs were protected against PCV2 infection (five pigs protected out of five) or against *M. hyorhinis* infection (four pigs out of five), concluding that their results demonstrate the effectiveness of this divalent experimental vaccine, which are similar to those of the present study (Table 8).

The experimental formulation proposed in this work uses ChS that behaves as a biomimetic of HS receptor, for encapsulation of antigens from pathogens that use HS as a cell receptor (US Patent 11246839B2) [36], which was not previously considered for vaccine development. The HS receptor plays an important role during adhesion and internalization processes of many strict and facultative intracellular pathogens, including bacteria and viruses [60,61,62,63,64,65].

The ChS biomimetic properties support the design of the vaccine prototype and is important for the microencapsulation process of antigens, improving their concentration by conjugating with ChS and facilitating formation of microparticles, as previously described [37]. Therefore, it is important to consider high degrees of deacetylation (>50%) and low-molecular-weight sulfated chitosan, as it has been previously optimized [37,66]. The experimental design used in this work is a modified version of the protocols previously described [67].

Immunization by the intramuscular route favors the humoral immune response (Th2), whereas the O/N route favors a mixed response (Th1/Th2) [12]. Pigs are considered immunocompetent at birth, but unable to develop a rapid response. Therefore antibodies-mediated protection relies on antibodies obtained though colostrum during the lactation period [68]. This event would explain the high levels of antibodies detected at early stages in this study and their rapid decrease in the absence of immune stimulation (see Figures on antibody titers). However, it must be considered that none of the approaches used detect the origin or isotype of the antibodies.

Passive immunity can affect the immunogenic effect of vaccines, mainly at early vaccination ages [69]. The mechanism of this phenomenon is unknown. This effect has been observed in vaccines for PCV2 and *M. hyopneumoniae* [70,71]. However, there are controversies between authors on this aspect [72,73], and to date, previous reports on this aspect of vaccines against *M. hyorhinis* are scarce.

Martinson et al. [17] demonstrated the efficacy of a new inactivated vaccine against *M. hyorhinis* for reducing of lameness and polyserositis in pigs. Subsequently, the same group [18] evaluated the same inactivated vaccine against *M. hyorhinis* at 7 or 10 weeks post-vaccination to determine the duration of the protective immune response. Efficacy of inactivated vaccine against *M. hyorhinis* using caesarian-derived, colostrum-deprived pigs was evaluated in animals vaccinated at 3 weeks of age and challenged with *M. hyorhinis* at either 10 weeks or 13 weeks of age. Results indicate duration of immunity of at least seven weeks after vaccination. In addition, the authors reported that no serological assays for *M. hyorhinis* were available at the time of this study; therefore, antibody titers were not detected for *M. hyorhinis*.

In the case of PCV2, both the commercial and the experimental vaccine described in the present study (considering both routes of administration) presented a significant difference compared with the placebo control. Therefore, evidence exists to conclude that the experimental formulation is effective against PCV2, acting in a similar way (in terms of antibodies induction) to the commercial vaccine, especially when administered via IM (Figure 2).

Regarding *M. hyopneumoniae*, the commercial vaccine (group 4) presented significant differences with the other groups (Groups 1, 2 and 3), being the only one showing an abrupt decrease of antibodies at T2 and T3, as seen in Table 8; the O/N route of the experimental formulation did not present positive individuals for *M. hyopneumoniae*, as also observed for the commercial vaccine, indicating restriction of colonization to this bacterium. This observation is important because of low antibody response against this bacterium with the group receiving the commercial vaccine. This may be evidence of decreased infection in the group that received the experimental formula via O/N (group 2), indicating that detection of serum antibodies (humoral immunity) for this bacterium is not the only indicator of immunocompetence in pigs against *M. hyopneumoniae*. There is sufficient evidence demonstrating that immunity dependent on serum antibodies is not always effective to combat respiratory infections in strict intracellular microbes compared to the cellular immune response [72,74,75].

Intramuscular administration of the experimental vaccine (group 1) presented only one positive individual for *M. hyopneumoniae*, compared with the placebo control (group 3), where three positive individuals were detected. This group behaved similar to group 4 (Table 8), whose control pigs received the commercial vaccine, a result that shows a protective activity of the experimental vaccine (IM route) against the primary causative agent of PRC.

Finally, for *M. hyorhinis* a significant increase in antibody titers was observed in the groups that received the experimental formula by both routes of administration (IM and O/N) with respect to the control groups (group 3 and 4), but differences were observed in terms of colonization of the lower respiratory tract (Table 8), which typically concurs with an imbalance of nasal microbiota in pigs, facilitating colonization beyond the upper respiratory tract [76]. The only group that did not present positive individuals for *M. hyorhinis* was the intramuscular administration of the experimental vaccine. This observation is important since no previous reports include *M. hyorhinis* in their experimental vaccines in association with PCV2 and *M. hyopneumoniae*.

*M. hyorhinis* frequently co-infects with various pathogens associated with respiratory diseases [5,77,78,79]. The coinfection of PCV2 and *M. hyorhinis* induces more severe lung lesions than those observed in pigs infected with *M. hyorhinis* or PCV2 alone [6], It has not been clearly identified to what extent this pathogen participates in increasing respiratory pathologies in pigs. In recent years, protecting pigs against *M. hyorhinis* has reduced severity of PRC [16]. Other authors describe that it is necessary to examine dual vaccination against PCV2 and *M. hyorhinis* experimentally to clarify the protective effect on PCV2 or *M. hyorhinis*-infected pigs [10]; however, research with experimental challenges is also relevant to evaluate the efficiency of trivalent vaccines (PCV2, *M. hyopneumoniae* and *M. hyorhinis*), as in the case of this study.

The above discussed results partially explain the promising results and effectiveness of the experimental vaccine (group 1) against three pathogens causing PRC. Importantly, it was observed that the route of administration of the experimental vaccine is a determining variable affecting the colonization of the respiratory tract by *M. hyopneumoniae*, *M. hyorhinis* and viral detection of PCV2 (Table 8). For this reason, in addition to discussing its influence on the immune response, we must also consider the pharmacokinetic aspect of the administration route of experimental formulations (compare groups 1 and 2). The IM administration route allows entry into the circulation systemically, increased bioavailability and possibly better tissue distribution [80] compared with the O/N route of administration.

ChS is biomimetic of cellular HS receptors, which is necessary for intracellular entry of the pathogens evaluated [23,36,37,81]. In vitro tests carried out in our laboratory demonstrated that ChS significantly decreased genomic copies, cellular infectivity titers and the PCV2 capsid protein, evidencing specific antiviral effects. Such effects depend on ChS molecular weight, concentration and chemical functionalization with S atoms [37]. Therefore, ChS would function as a decoy molecule due to similarities with the HS receptor, contributing to reduce the opportunity of pathogens to bind to the authentic HS receptor and consequently reducing the level of infected (permissive) target cells. Such effects could operate better in pigs of group 1 (possibly due to IM administration route) by restricting or blocking the infections of primary CRP pathogens, with respect to the rest of the experimental groups.

In addition, chitosan is also used as a drug vehicle and adjuvant [21,28], and it is suggested that the functionalization of chitosan with sulfur atoms (sulfonation) may contribute to increase the lipid solubility of this experimental formulation. The ChS could further increase bioavailability and tissue distribution, which has been reported in anthelmintic drug molecules for veterinary use, as sulfur atoms were incorporated [82]. It is important to mention that no side effects were observed at the injection site in the case of the injectable formulation or mucous membrane inflammation in the oro/nasal formula. This makes ChS-based microparticles a safe vehicle for animal vaccination.

Finally, this study is the first report worldwide which describes the analysis of a multivalent experimental vaccine through a challenge involving the inoculation of three pathogens (PCV2 and *M. hyopneumoniae* and *M. hyorhinis*), with its antigens microencapsulated with functionalized chitosan. Although the inoculum for the challenge (obtained from a lung macerate) was tested as negative for porcine reproductive and respiratory syndrome virus (PRRSV), swine influenza virus (SIV), Porcine Parvovirus and *Actinobacillus pleuropneumoniae*, it may have contained other not-detected respiratory pathogens that may have interfered with the proper interpretation of the results.

Finally, the results obtained validate an innovative biotechnological strategy in veterinary medicine to contribute to the prevention and control of PRC with a One Health approach, restricting or reducing the use of antimicrobials in PRC therapy, and reducing the risk of antimicrobial resistance and its economic costs.

## 5. Conclusions

This study is the first proposing to evaluate an experimental vaccine that includes *M. hyorhinis* with PCV2 and *M. hyopneumoniae*, whose antigens were microencapsulated with sulfated chitosan, in addition to establishing the first in vivo test model in pigs worldwide that allows immunizing and challenging with three pathogens simultaneously.

The physicochemical evaluation of the formulation with MPs of ChS + PRC-antigens had a size between 1.3–10 μm showed a low Pdi, while the zeta potential was negative (−33.2 mV) with respect to the MPs of Chc (50.3 mV), probably due to the higher concentration of S (4.26%) compared to MPs of Chc + PRC-antigens (0.02%).

Intramuscular (IM) immunization was more effective with a lower positivity for PCV2 compared with the oral/nasal route and the commercial vaccine. In the case of *M. hyopneumoniae*, the experimental formulation and the commercial vaccine performed similarly according to a postmortem analysis (PCR-Real Time test) of bacterial load.

Finally, the experimental vaccine induced a significant increase in antibody titers for *M. hyorhinis* by both routes of administration, compared with the commercial vaccine. However, analysis (PCR-Real Time test) demonstrated that only intramuscular administration of the experimental formulation was effective in preventing colonization of the lower respiratory tract. This observation is very important because there are no vaccine alternatives against *M. hyorhinis* associated with PCV2 and *M. hyopneumoniae*.

These results demonstrate that it is feasible to use MPs with ChS as a simple and effective system for the administration of PCV2, *M. hyopneumoniae* and *M. hyorhinis* antigens. Thus, intramuscular and oral/nasal administration may be an effective strategy to induce protection and antibody titers against these PRC pathogens.

## Figures and Tables

**Figure 1 vaccines-12-00550-f001:**
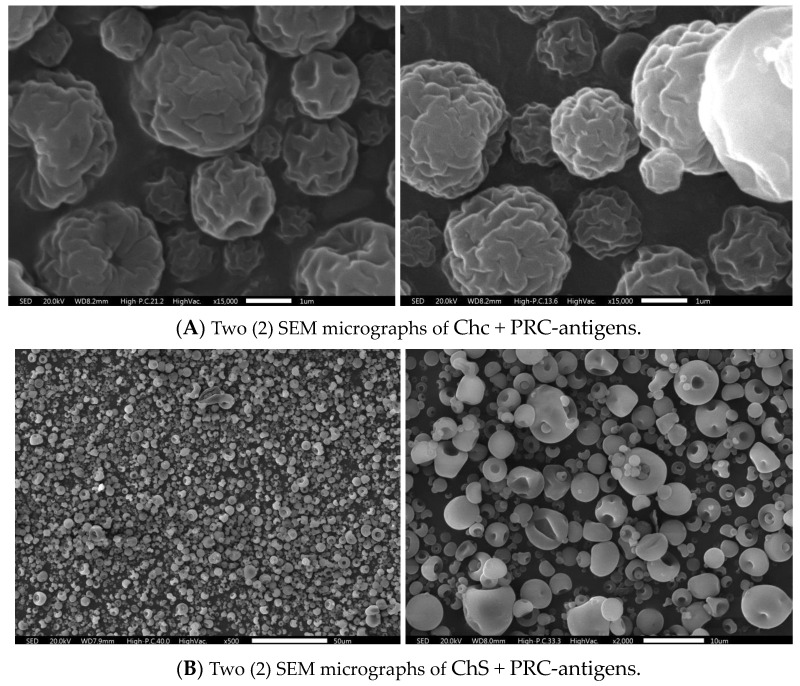
Scanning electron microscopy (SEM). (**A**) shows two micrographs. The particles of Chc (commercial chitosan) + PRC-antigens (porcine respiratory complex antigens), which are spherical particles of 1.2–6.5 μm of irregular surface with observed concavities and grooves. (**B**) shows two micrographs. The particles of ChS (sulfated chitosan) + PRC-antigens are mainly spherical morphology with smooth surface concavities and rough surface, which have approximate sizes of 1.3–10 μm.

**Figure 2 vaccines-12-00550-f002:**
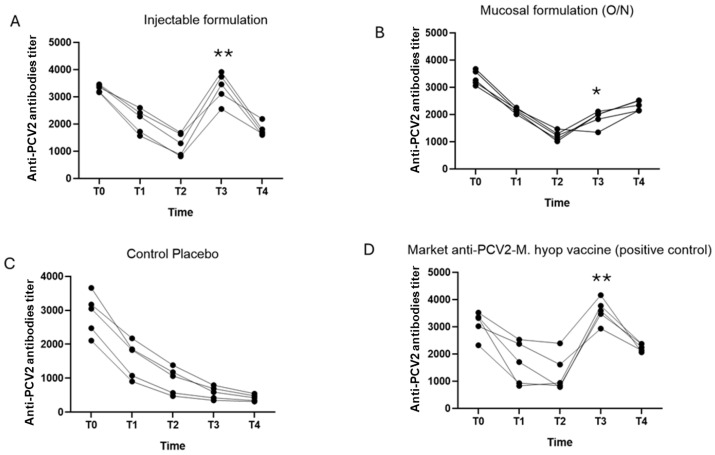
Induction of antibodies against PCV2. (**A**) Graph of the antibody titers for Group 1 injectable formulation (Experimental formula IM). (**B**) Graph of antibody titers for Group 2 mucosal formulation (Experimental formula O/N). (**C**) Graph of the antibody titers for Group 3 (control placebo). (**D**) Graph of antibody titers for Group 4 commercial vaccine (control positive). The antibody titer was expressed as the total number of antibodies. The symbol * indicates significant difference (* *p* ≤ 0.05 and ** *p* ≤ 0.01) between treatments.

**Figure 3 vaccines-12-00550-f003:**
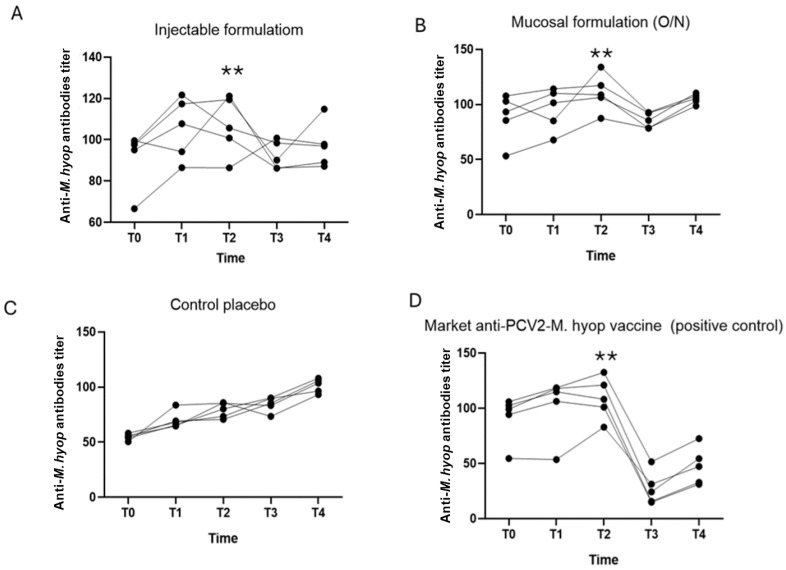
The results of antibody induction for *M. hyopneumoniae* (*M. hyop*). (**A**) Graph of the antibody titers for Group 1 injectable formulation (Experimental formula IM). (**B**) Graph of antibody titers for Group 2 mucosal formulation (Experimental formula O/N). (**C**) Graph of the antibody titers for Group 3 (control placebo). (**D**) Graph of antibody titers for Group 4 commercial vaccine (control positive). The antibody titer was expressed as the total number of antibodies. The symbol ** indicates results present significant differences (*p* ≤ 0.01) between the different treatments.

**Figure 4 vaccines-12-00550-f004:**
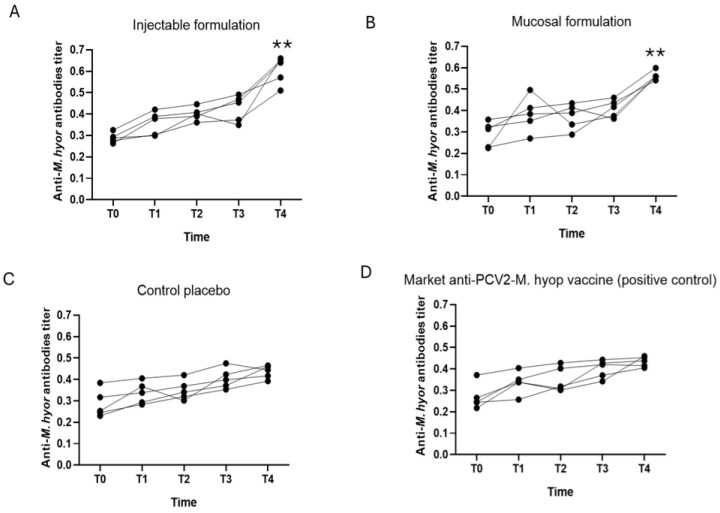
Summary of the results of the induction of antibody titers for *M. hyorhinis* (*M. hyor*). (**A**) Graph of the antibody titers in Group 1 injectable formulation (Experimental formula IM). (**B**) Graph of antibody titers for Group 2 mucosal formulation (Experimental formula O/N). (**C**) Graph of the antibody titers in Group 3 (control placebo). (**D**) Graph of antibody titers for Group 4, commercial vaccine (control positive). The titer of antibodies is expressed as the value of the optical density (OD) obtained at 450 nm. The symbol ** indicates the results where of significant difference (*p* ≤ 0.01) between different treatments.

**Table 1 vaccines-12-00550-t001:** Groups of pigs exposed to the 4 experimental treatments and administered volume of each formulation.

Group	Number of Animals	Experimental Treatment
1	5	Experimental formula 1 mL (1 mg/mL of MPs)–IM
2	5	Experimental formula 3 mL (1 mg/mL of MPs)–O/N
3	5	Placebo–3 mL (1 mg/mL of MPs)–O/N
4	5	Commercial vaccine 1 mL (1 dose)–IM

Group 1: microparticles lyophilized of sulfated chitosan + antigens (PCV2, *M. Hyopneumoniae* and *M. Hyorhinis*) suspended in PBS (1 mL; final concentration 1 mg/mL). Group 2: the same formulation of group 1, but was administered orally (2 mL) and nasally (1 mL). Group 3: control (placebo) formulation with MPs of sulfated chitosan (without pathogenic antigens). Group 4: commercial vaccination (vaccine Suvaxyn^®^ Gold PCV MH). Route of administration IM: intramuscular; O/N: oral/nasal (MPs: lyophilized microparticles).

**Table 2 vaccines-12-00550-t002:** Description of the experimental period of the research (12 weeks), indicating the event, procedure, time (T) experiment, the period in which it occurred and whether sampling and type of samples were performed.

Event and/or Experimental Procedure	Day of the Event in Experimental Period	Duration of the Period (Days)	Number of Week	Sample
Time 0 (Acclimatization)	1	10	1,2	nasal swab/blood
Time 1/first immunization and start of the experiment (administration of experimental formulations, placebo and commercial vaccine for each treatment)	11	14	2,3	nasal swab/blood
Time 2	25	7	4	nasal swab/blood
Vaccine boosters (21 days after the 1st immunization)	31	7	5	-
Time 3	38	7	6	nasal swab/blood
Experimental challenge	45	7	7	-
Time 4/Vaccine boosters (42 days after the 1st immunization)	52	7	8	nasal swab/blood
Time 5	68	16	9,10,11	nasal swab/blood
Time 6 (necropsy)	80	12	11,12	Bronchial swab/inguinal lymph node/blood

**Table 3 vaccines-12-00550-t003:** List of primers and probes used during this study.

Primers	Sequence
Fw PCV2	5′-GGGATGATCTACTGAGACTGTGTGA-3′
Rv PCV2	5′-GGGGAAAGGGTGACGAACT-3′
Probe PCV2	5′-/56-FAM/AATGGTACT/ZEN/CCTCAACTGCTGTCCCAGC/3IABkFQ/-3′
Fw *M. hyop*	5′-GTCAAAGTCAAAGTCAGCAAAC-3′
Rv *M. hyop*	5′-AGCTGTTCAAATGCTTGTCC-3′
Probe *M. hyop*	5′-/5Cy5/ACCAGTTTC/TAO/CACTTCATCGCCTCA/3IAbRQSp/-3′
Fw *M. hyor*	5′-CGGGATGTAGCAATACATTCAG-3′
Rv *M. hyor*	5′-AGAGGCATGATGATTTGACGTC-3′

**Table 4 vaccines-12-00550-t004:** Detection of AND of PCV2, *M. hyopneumoniae* and *M. hyorhinis*.

	PCV2/*M. hyopneumoniae*	*M. hyorhinis*
Method	Real Time qPCR Multiplex	Real Time PCR (Sybr Green)
Temperature	95 °C	95 °C
Time	3 min	3 min
Cycles	40 (95 °C for 15 s and 60 °C per 1 min)	40 (95 °C for 15 s and 60 °C for 35 s)

**Table 5 vaccines-12-00550-t005:** Energy Dispersive Spectroscopy (EDS).

Element	Concentration (%)	
	ChS + PRC-Antigens	Chc + PRC-Antigens
C	41.35	68.01
N	9.68	6.81
O	44.4	25.17
S	4.56	0.02
Total	100	100

ChS (sulfated chitosan); Chc (commercial chitosan); PRC-antigens (porcine respiratory complex antigens).

**Table 6 vaccines-12-00550-t006:** Analysis of Surface Charge and Polydispersity of Microparticles (MPs).

MPs + Antigen	Zeta Potential (mV)	Polydispersity (PdI)
Chc + *M*. *hyorhinis*	48.2	0.595
ChS + *M*. *hyorhinis*	−21.7	0.314
Chc + *M*. *hyopneumoniae*	44.7	0.712
ChS + *M*. *hyopneumoniae*	−38.8	0.210
Chc + PCV2	58.2	0.680
ChS + PCV2	−39.1	0.293

ChS (sulfated chitosan); Chc (commercial chitosan); PdI (polydispersity index).

**Table 7 vaccines-12-00550-t007:** PCR-Real Time results showing Ct (cycle threshold) values for each animal (ID), obtained from postmortem bronchial (PCV2, *M. hyopneumoniae and M. hyorhinis*) and inguinal lymph node (PCV2) samples (T6). The minus sign (−) indicates no detection by the equipment.

		Samples Obtained in the Post-Mortem Experimental Period (T6) during the Necropsy				
		Bronchial Swab			Inguinal Lymph Node	Sera
	Pathogen	PCV2	*M. hyop.*	*M. hyor.*	PCV2	PCV2
Animal (ID)						
1		−	−	−	+	−
2		−	+	+	+	−
3		+	−	−	+	−
4		+	−	+	+	−
5		−	−	−	−	−
6		+	−	+	−	−
7		−	−	+	+	−
8		−	−	−	−	−
9		+	+	+	+	−
10		−	+	+	+	−
11		−	−	−	−	−
12		+	−	−	+	−
13		−	−	−	+	−
14		+	−	+	−	−
15		−	−	+	−	−
16		−	−	+	+	−
17		+	+	−	+	−
18		−	−	−	+	−
19		+	−	+	−	−
20		−	−	−	−	−

Sample data shown from T0 to T5 were used to monitor the presence of pathogens throughout the study. Importantly, T6 results account for the final state of the animals for each group of pathogens in the study. The bronchial swab showed if there was colonization in the lower respiratory tract of the individuals, while the lymph node sample indicated whether PCV2, as mentioned above, was capable of infecting the lymphatic system. Finally, the blood sample shows if there was viremia in the individuals. (ID: identification code.)

**Table 8 vaccines-12-00550-t008:** The total number of positive individuals divided by the total n in each experimental group (positive by Real Time PCR) for PCV2, *M. hyopneumoniae* and *M. hyorhinis*, sampling was carried out post mortem at sampling time 6 (T6).

	Pathogen	PCV2	*M. hyopneumoniae*	*M. hyorhinis*
Group				
(1) Experimental formula (IM)		1/5	1/5	0/5
(2) Experimental formula (O/N)		5/5	0/5	3/5
(3) Placebo control (O/N)		3/5	3/5	3/5
(4) Commercial vaccine control (IM)		3/5	0/5	3/5

The table summarizes the data in Table 7, which shows that group 3 (placebo) tested 60% of its individuals positive for each pathogen. The commercial vaccine (group 4) had the same percentage of PCV2- and *M. hyor*-positive individuals, but was able to protect 100% of the group against M. hyop (*M. hyopneumoniae*). The experimental formulation administered intramuscularly (group 1) protected 80% of its individuals against PCV2 and *M. hyop*, and was the only treatment that protected 100% against *M. hyor* (*M. hyorhinis*). Finally, the experimental group via oro-nasal (group 2) presented 100% of its individuals positive for PCV2. For *M. hyor* the behavior was similar to control (group 4), while for *M. hyop* it protected 100% of its individuals. Route of administration IM: intramuscular; O/N: oral/nasal; MPs: microparticles. Group 1: Experimental formula 1 mL (1 mg/mL of MPs)—IM. Group 2: Experimental formula 3 mL (1 mg/mL of MPs)—O/N. Group 3: Placebo control–3 mL (1 mg/mL of MPs)—O/N. Group 4: Commercial vaccine control–1 mL (1 dose)—IM.

## Data Availability

The datasets used and/or analyzed during the current study are available from the corresponding author on reasonable request.

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
