# Peer review of "Design of a New Vaccine Prototype against Porcine Circovirus Type 2 (PCV2), *M. hyopneumoniae* and *M. hyorhinis* Based on Multiple Antigens Microencapsulation with Sulfated Chitosan"

_vaccines, 2024, doi:10.3390/vaccines12050550_

Round 1

Reviewer 1 Report

Comments and Suggestions for Authors

1.  Title: correct

2.  Abstract: contains unexplained abbreviations ( EDS, Pdi, S) – please correct;

3.  Introduction: contains incorrect information about PCV2:

- PMWS is the outdated name of one of the form of the disease linked to PCV2 (line 58); also, more recent citation should be mentioned here;

- the infection with this virus does not have to result in lesions (compare line 59) or “significant depression of the immune system” (line 62).

4.  Materials and Methods:

- There is no information about the approval of the Ethics Committee;

- There is no information about the vaccination programme of the herd from which pigs used in the experiment were born (including vaccination against PCV2 clinical symptoms, or M. hyopneumoniae);

- there is no information about possible prior problems (clinical symptoms, necropsy lesions, results of laboratory tests) with PCV2, or Mycoplasmas, or other symptoms/agents (including PRDC), in the parental herd

-  it is not clear if pigs from experimental groups were located in different pens, or in the same one

- pigs were divided into groups based on similar weight, so it is not true that they were randomly distributed (line 127)

- table 1: below the table there is repetition of the information from the main text

- the anatomical name of the muscle seems incorrect (line 195)

- subunit “Experimental challenge” contains no information about the inoculum: the source of microorganisms, their amount, etc.; also, the sentence needs to be corrected: “Eight weeks after starting inoculation…” (incorrect word, line 203)

- RT-PCR” refers to “Reverse Transcription Polymerase Chain Reaction” which has no use in case of microorganisms containing DNA; the Authors probably meant real-time PCR – please correct;

-   The sentence in lines 208-210 is unclear;

- line 222: the sentence “After inoculation, the animals were observed until complete recovery” – did you mean until the pigs were observed until sedatives stopped working? Please specify, because in the present form it suggests you expected all the pigs to be ill after inoculation;

- line 239: “AVMA” should be first mentioned with full name, then used as an abbreviation;

- the sentence starting in line 239: “The samples …(…)” is unclear;

- PCR details (line 256-261) should rather be presented in a separate table

- Line 270-273: these 2 sentences about M. hyorhinis ELISA should be combined into 1;

- source of serum (line 277), secondary antibody (line 278) and  BSA (line 282) should be provided;

-  line 290 and 298: “to reveal the plate” is a colloquial term that should be replaced with a correct one;

5.       Results: - Table 6B needs English correction;

               - Table 6B description does not mention in which tests individuals were positive for each pathogen;

- line 355 – 356: this is not true since the animals had antibodies which means they must have had prior contact with pathogens; also, the parental herd status is not mentioned, so the situation is unknown;

-   line 362: lungs should be also checked for PCV2 due to the involvement of this virus in PRDC;

-  line 379-380: missing words;

-   line 386-387: unclear sentence;

-   sentence in lines 414-416 needs correction;

-   Fig. 3 and Fig. 4 need English correction (all graphs);

- it is surprising why the Authors did not perform/mention the results of necropsy of experimental pigs;

- also, neither histopathological analysis nor in situ detection of PCV2 were performed – both would clearly prove that inoculation with PCV2 did not result in any clinical form of this infection;

-   line 414: needs correction

6.       Discussion: fine, but too long;

7.       Conclusions: the explanation concerning similar titer of antibodies against M. hyorhinis does not seem to be convincing since all the tested microorganisms are ubiquitous and widespread in swine (lines 738-741);

8.       References: too many old – and, in many cases, outdated papers cited.

Comments on the Quality of English Language

The manuscript needs English correction because it contains some incorrect words, some words (or even parts of sentences) are missing, some words have  not been translated into English.

Author Response

Dear Editor,

Please find enclosed our responses to the reviewer’s comments for the manuscript “Titled: “Design of a New Vaccine Prototype Against Porcine Circovirus Type 2 (PCV2), M. hyopneumoniae and M. hyorhinis Based on Multiple Antigens Microencapsulation with Sulfated Chitosan”.
by: Darwuin Arrieta-Mendoza, Bruno Garces, Alejandro Hidalgo, Victor
Neira Ramirez, Galia Ramirez, Andrónico Neira-Carrillo, Sergio A Bucarey. The authors appreciate all comments provided by the referees, which have greatly helped us in reviewing our article and highlight the main aim of our research. Therefore, we feel that this new version of the manuscript is much clear and highlight the novelty of our scientific work.

All changes made in the text are in accordance with the reviewer's comments. Our point-by-point responses to each of the reviewer's comments are included below and are denoted in italics.

We believe our manuscript has been significantly improved by introducing the above-mentioned modification. We really appreciate your valuable opinion and guidance.

Sincerely yours,

Corresponding Authors

Sergio A. Bucarey

Response to Reviewers Comments

Reviwer 1

Comments and Suggestions for Authors

  1. Title: correct

Response:  We thank the reviewer for this observation.

  1. Abstract: contains unexplained abbreviations ( EDS, Pdi, S) – please correct;

Response: It is done. We thank the reviewer for this observation.

  1. Introduction: contains incorrect information about PCV2:

- PMWS is the outdated name of one of the form of the disease linked to PCV2 (line 58); also, more recent citation should be mentioned here;

Response: We thank the reviewer for this valuable observation, which was incorporated into the manuscript.

- the infection with this virus does not have to result in lesions (compare line 59) or “significant depression of the immune system” (line 62).

Response: We thank the reviewer for this valuable observation, which was incorporated into the manuscript.

  1. Materials and Methods:

- There is no information about the approval of the Ethics Committee;

Response: We thank the reviewer for this valuable observation, which was incorporated into the manuscript.

- There is no information about the vaccination programme of the herd from which pigs used in the experiment were born (including vaccination against PCV2 clinical symptoms, or M. hyopneumoniae);

Response: We thank the reviewer for this valuable observation, which was incorporated into the manuscript, providing the corresponding information

- there is no information about possible prior problems (clinical symptoms, necropsy lesions, results of laboratory tests) with PCV2, or Mycoplasmas, or other symptoms/agents (including PRDC), in the parental herd

Response: We thank the reviewer for this valuable observation, which was incorporated into the manuscript, providing the corresponding information.

-  it is not clear if pigs from experimental groups were located in different pens, or in the same one

Response: We thank the reviewer for this valuable observation, which was incorporated into the manuscript. The pigs were placed separately according to the experimental group.

- pigs were divided into groups based on similar weight, so it is not true that they were randomly distributed (line 127).

Response: We thank the reviewer for this valuable observation, which was clarified and incorporated into the manuscript. Specifying that the pigs were randomly distributed into 4 groups.

- table 1: below the table there is repetition of the information from the main text

Response: We thank the reviewer for this valuable observation, which was clarified and incorporated into the manuscript. Specifying that the pigs were randomly distributed into 4 groups. Only placing the necessary information in the table legend.

- the anatomical name of the muscle seems incorrect (line 195)

Response: We thank the reviewer for this valuable observation, which was clarified and incorporated into the manuscript.

- subunit “Experimental challenge” contains no information about the inoculum: the source of microorganisms, their amount, etc.; also, the sentence needs to be corrected: “Eight weeks after starting inoculation…” (incorrect word, line 203).

Response: We thank the reviewer for this valuable observation, which was incorporated into the manuscript. With more information on the inoculum used and the indicated word was changed and used.

 - ”RT-PCR” refers to “Reverse Transcription Polymerase Chain Reaction” which has no use in case of microorganisms containing DNA; the Authors probably meant real-time PCR – please correct;

Response: We thank the reviewer for this valuable observation, which was incorporated into the manuscript.

-   The sentence in lines 208-210 is unclear;

Response: It is done. We thank the reviewer for this observation. improving text writing

- line 222: the sentence “After inoculation, the animals were observed until complete recovery” – did you mean until the pigs were observed until sedatives stopped working? Please specify, because in the present form it suggests you expected all the pigs to be ill after inoculation;

Response: We thank the reviewer for this valuable observation, which was incorporated into the manuscript. Explaining that they waited for the recovery of the animals after sedation.

- line 239: “AVMA” should be first mentioned with full name, then used as an abbreviation;

Response: We thank the reviewer for this observation, which was incorporated into the manuscript.

- the sentence starting in line 239: “The samples …(…)” is unclear;

Response: It is done. We thank the reviewer for this observation. improving text writing

- PCR details (line 256-261) should rather be presented in a separate table

Response: We thank the reviewer for this observation, Following the suggestion, the information was organized in a table (table 3a)

- Line 270-273: these 2 sentences about M. hyorhinis ELISA should be combined into 1;

Response: It is done, improving text writing. We thank the reviewer for this observation.

- source of serum (line 277), secondary antibody (line 278) and  BSA (line 282) should be provided;

Response: It is done. We thank the reviewer for this observation.

-  line 290 and 298: “to reveal the plate” is a colloquial term that should be replaced with a correct one;

Response: It is done, improving text writing. We thank the reviewer for this observation.

  1. Results:

- Table 6B needs English correction;

Response: It is done, improving the english of the text. We thank the reviewer for this observation.

  - Table 6B description does not mention in which tests individuals were positive for each pathogen;

Response: Thank you for your kind and valuable reminder. This information is described in table 6, at the beginning of the first title: "B) Summary of the total number of individuals divided by experimental group that were positive (PCR-Real Time test) for PCV2, M. hyopneumoniae and M. hyorhinis, at the end of the experiment"... Additionally, in the legend of table 6B, it was indicated that: "table summarizes the data in table 6A"

- line 355 – 356: this is not true since the animals had antibodies which means they must have had prior contact with pathogens; also, the parental herd status is not mentioned, so the situation is unknown;

Response:  The authors appreciate this important comment. We recognize the need for the suggested information on this point. Therefore, the corresponding information was incorporated into the manuscript about the parental herd, to expand this paragraph at this point of reflection on the results.

-   line 362: lungs should be also checked for PCV2 due to the involvement of this virus in PRDC;

Response: We thank the reviewer for this comment. It is true, it is another type of very useful sample in the diagnosis of PCV2.

-  line 379-380: missing words;

Response: It is done. We thank the reviewer for this observation.

-   line 386-387: unclear sentence;

Response: It is done, improving the text. We thank the reviewer for this observation.

-   sentence in lines 414-416 needs correction;

Response: It is done, improving text writing. We thank the reviewer for this observation.

-   Fig. 3 and Fig. 4 need English correction (all graphs);

Response: It is done. We thank the reviewer for this observation, improving the english of the text of graphs.

- it is surprising why the Authors did not perform/mention the results of necropsy of experimental pigs;

Response: The authors appreciate this important comment. We are aware of the important information provided by the suggested analysis; however, this type of study was not carried out in this preliminary experiment, for logistical and financial reasons. We will incorporate this information in a second part of the investigation.

- also, neither histopathological analysis nor in situ detection of PCV2 were performed – both would clearly prove that inoculation with PCV2 did not result in any clinical form of this infection;

Response: We thank the reviewer for this observation. In the same order of ideas, from the previous comment. We will incorporate this information in a second part of the investigation. We are aware of the important information provided by the suggested analysis; however, this type of study was not carried out in this preliminary experiment, for logistical and financial reasons.

-   line 414: needs correction

Response: It is done. We thank the reviewer for this observation.

  1. Discussion: fine, but too long;

Response: The authors appreciate the observation, so some lines were eliminated and paragraphs were modified.

  1. Conclusions:

- the explanation concerning similar titer of antibodies against M. hyorhinis does not seem to be convincing since all the tested microorganisms are ubiquitous and widespread in swine (lines 738-741);

Response: We thank the reviewer for this valuable observation, which we reflected on and modified to have an appropriate conclusion.

  1. References:

- too many old – and, in many cases, outdated papers cited.

Response: The authors appreciate this valuable observation. In this regard, some references were removed, and more updated ones were incorporated into the manuscript.

Comments on the Quality of English Language

The manuscript needs English correction because it contains some incorrect words, some words (or even parts of sentences) are missing, some words have  not been translated into English.

Response: The authors appreciate this valuable observation, and an English revision was carried out on the manuscript.

Reviewer 2 Report

Comments and Suggestions for Authors

The authors of this manuscript evaluated an experimental multivalent vaccine targeting Porcine Respiratory Complex (PRC)-associated antigens: Porcine Circovirus Type 2 (PCV2), M. hyopneumoniae (Mhyop), and M. hyorhinis (Mhyor). The vaccine was microencapsulated with sulfated chitosan to mimic heparan sulfate, a receptor used by these pathogens for cell invasion. Physicochemical analysis using SEM, EDS, PDI, and zeta potential measurements showed that the MEV particles ranged in size from 1.3 to 10 μm, displaying low PDI values and negative zeta potential, likely due to the presence of sulfur content. The animal study showed that intramuscular immunization was more effective against PCV2 compared to oral/nasal routes and a commercial vaccine. The experimental formulation produced a lower humoral response for M. hyopneumoniae but reduced bacterial load similarly. For M. hyorhinis, similar antibody titers were observed in all groups, but intramuscular administration effectively prevented lower respiratory tract colonization. These results suggest that microencapsulated particles with sulfated chitosan could be an effective system for administering antigens against these pathogens.

However, a significant limitation of this study is the lack of efforts to reduce the interference of preexisting exposure to pathogens and maternal antibodies. Screening animal candidates with PCR for pathogen DNA and using ELISA for antibody detection could enhance the reliability of the study results. Selecting an appropriate age for the animals also helps to reduce the impact of maternal antibodies. Additionally, several concerns need to be addressed.

1. Line 20, “experimental-multivalent-vaccine (MEV) based” appears not accurate. Did you mean "microencapsulated vaccine (MEV)"? Forethermore, instead of using the abbreviation MEV only in the abstract, a precise and consistent term should be used throughout the manuscript.

2. The use of "QS" as an abbreviation for chitosan sulfate is non-standard. Standard abbreviations should be used in the manuscript, and terms should be clearly defined. For instance, instead of "BSA-T," it would be clearer to use "PBS-T with 3% BSA" or to define  it as "blocking buffer" after its components are initially listed.

3. Table 1, which repeats text contents, is unnecessary and can be omitted.

4. Table 2 needs clarification, particularly regarding the timing of the vaccine booster shot after the challenge. Time points in the table should be precise and consistent with the flow of the full text. Abbreviations of events described in the table should not be used to present results. The table should use days as the time unit, starting from the beginning of the experiment.

5. To ensure logical coherence, the analysis of vaccine particles should be described before the animal study in the Materials and Methods section.

6. The term "positive" in qPCR should be clearly defined in the manuscript.

7. References should be cited properly. For example, reference 37 lacks a source citation and needs to be corrected accordingly, please check all references carefully.

Author Response

Dear Editor,

Please find enclosed our responses to the reviewer’s comments for the manuscript “Titled: “Design of a New Vaccine Prototype Against Porcine Circovirus Type 2 (PCV2), M. hyopneumoniae and M. hyorhinis Based on Multiple Antigens Microencapsulation with Sulfated Chitosan”.
by: Darwuin Arrieta-Mendoza, Bruno Garces, Alejandro Hidalgo, Victor
Neira Ramirez, Galia Ramirez, Andrónico Neira-Carrillo, Sergio A Bucarey. The authors appreciate all comments provided by the referees, which have greatly helped us in reviewing our article and highlight the main aim of our research. Therefore, we feel that this new version of the manuscript is much clear and highlight the novelty of our scientific work.

All changes made in the text are in accordance with the reviewer's comments. Our point-by-point responses to each of the reviewer's comments are included below and are denoted in italics.

We believe our manuscript has been significantly improved by introducing the above-mentioned modification. We really appreciate your valuable opinion and guidance.

Sincerely yours,

Corresponding Authors

Sergio A. Bucarey

Response to Reviewers Comments

Reviwer 2

Comments and Suggestions for Authors

The authors of this manuscript evaluated an experimental multivalent vaccine targeting Porcine Respiratory Complex (PRC)-associated antigens: Porcine Circovirus Type 2 (PCV2), M. hyopneumoniae (Mhyop), and M. hyorhinis (Mhyor). The vaccine was microencapsulated with sulfated chitosan to mimic heparan sulfate, a receptor used by these pathogens for cell invasion. Physicochemical analysis using SEM, EDS, PDI, and zeta potential measurements showed that the MEV particles ranged in size from 1.3 to 10 μm, displaying low PDI values and negative zeta potential, likely due to the presence of sulfur content. The animal study showed that intramuscular immunization was more effective against PCV2 compared to oral/nasal routes and a commercial vaccine. The experimental formulation produced a lower humoral response for M. hyopneumoniae but reduced bacterial load similarly. For M. hyorhinis, similar antibody titers were observed in all groups, but intramuscular administration effectively prevented lower respiratory tract colonization. These results suggest that microencapsulated particles with sulfated chitosan could be an effective system for administering antigens against these pathogens.

However, a significant limitation of this study is the lack of efforts to reduce the interference of preexisting exposure to pathogens and maternal antibodies. Screening animal candidates with PCR for pathogen DNA and using ELISA for antibody detection could enhance the reliability of the study results. Selecting an appropriate age for the animals also helps to reduce the impact of maternal antibodies. Additionally, several concerns need to be addressed.

Response: The authors appreciate these important observations and reflections. In this regard, we will comment on the high cost of maintaining pigs to an age to overcome maternal antibodies.

Therefore, colostrum-deprived animals could be used as another alternative and from SPF mothers. However, it is also a financial cost that is not easy to meet at the moment, but will be considered for future research.

  1. Line 20, “experimental-multivalent-vaccine (MEV) based” appears not accurate. Did you mean "microencapsulated vaccine (MEV)"? Forethermore, instead of using the abbreviation MEV only in the abstract, a precise and consistent term should be used throughout the manuscript.

Response: It is done. We thank the reviewer for this observation.

  1. The use of "QS" as an abbreviation for chitosan sulfate is non-standard. Standard abbreviations should be used in the manuscript, and terms should be clearly defined. For instance, instead of "BSA-T," it would be clearer to use "PBS-T with 3% BSA" or to define  it as "blocking buffer" after its components are initially listed.

Response: It is done. We thank the reviewer for this observation.

  1. Table 1, which repeats text contents, is unnecessary and can be omitted.

Response: It is done. We thank the reviewer for this observation.

  1. Table 2 needs clarification, particularly regarding the timing of the vaccine booster shot after the challenge. Time points in the table should be precise and consistent with the flow of the full text. Abbreviations of events described in the table should not be used to present results. The table should use days as the time unit, starting from the beginning of the experiment.

Response: It is done. We thank the reviewer for this observation.

  1. To ensure logical coherence, the analysis of vaccine particles should be described before the animal study in the Materials and Methods section.

Response: It is done. We thank the reviewer for this observation.

  1. The term "positive" in qPCR should be clearly defined in the manuscript.

Response: It is done. We thank the reviewer for this observation.

  1. References should be cited properly. For example, reference 37 lacks a source citation and needs to be corrected accordingly, please check all references carefully.

Response: We thank the reviewer for this observation. the review of the references in the manuscript was performed and It is important to clarify that this reference refers to the patent on sulfated chitosan used in this research. The Zotero program was used to cite the references. In the Zotero program this reference was introduced in the patent option, this is probably the reason for the difference with the rest of the references. Additionally we must inform that the patent was added in section 6 of the manuscript (supplementary material).

Round 2

Reviewer 1 Report

Comments and Suggestions for Authors

Thank you for your extensive correction of this manuscript. I really appreciate the work you did. I accept these changes and I am grateful for the clarification of some issues.

As I suggested previously, this article would have greater scientific value with more results from other organs collected from experimental animals, but I accept and understand your explanation concerning logistic and financial reasons. Hopefully, these analyses will be available and can be published in future.

However, there are still some necessary corrections to be done. 

Please add to you results (at least) small paragraph about the condition of experimental animals. You mention physical examination of pigs (line 222), but there are no results commenting this. You should inform if there were any problems with pigs’ health during the course of experiment, with the special emphasis on pigs’ status after the challenge (any clinical symptoms, local reactions at the site of vaccine administration, etc.).

Also, you used lungs homogenate as inoculum, so since these organs were collected from the diseased pigs, you should also expect presence of other respiratory tract pathogens (viral, bacterial) in them. Therefore, I suggest to correct the sentence starting in the line 244: “In week 7 after starting the experimental treatments (T1), the challenge was carried out by inoculation the four study groups (table 2) with lung homogenate as a source of M. hyopneumoniae, M. hyorhinis and PCV2.” Consequently, please mention in the Discussion that the inoculum has not been tested for the presence of other pathogens (or please correct me if I’m wrong), and, most probably, it contained a “cocktail” of microorganisms. All of these must have influenced the results.

Comments on the Quality of English Language

1.   Please correct typo errors:

·         line 279: “lympnonode”

·         line 401: “undercontrolled”

·         line 403: “Threfore”

·         line 411: please use “animals” instead of “animal”

·         line 472: “titerss”

·         line 781: “fr”

2.       line 428: please replace “lymph node” with “lymphoid”

3.       line 485: please use “presenting” instead of “present”

4.       Please correct fig. 2 and 3: you still have “titles” there…

5.       Please remove unnecessary commas in lines: 60, 589, 606.

6.       Please check again the whole text for any other editioral mistakes.

Author Response

April  15th, 2024

Ms. Tessa Wang

Assistant Editor

MDPI Vaccines Editorial Office

Ref. Manuscript ID:  vaccines-2919578

Dear Editor,

Please find enclosed our responses to the reviewer’s comments for the manuscript “Titled: “Design of a New Vaccine Prototype Against Porcine Circovirus Type 2 (PCV2), M. hyopneumoniae and M. hyorhinis Based on Multiple Antigens Microencapsulation with Sulfated Chitosan”.
by: Darwuin Arrieta-Mendoza, Bruno Garces, Alejandro Hidalgo, Victor
Neira Ramirez, Galia Ramirez, Andrónico Neira-Carrillo, Sergio A Bucarey. The authors appreciate all comments provided by the referees, which have greatly helped us in reviewing our article and highlight the main aim of our research. Therefore, we feel that this new version of the manuscript is much clear and highlight the novelty of our scientific work.

All changes made in the text are in accordance with the reviewer's comments. Our point-by-point responses to each of the reviewer's comments are included below and are denoted in italics.

We believe our manuscript has been significantly improved by introducing the above-mentioned modification. We really appreciate your valuable opinion and guidance.

Sincerely yours,

Corresponding Authors

Sergio A. Bucarey

Response to Reviewers Comments

Reviwer 1

Comments and Suggestions for Authors

Thank you for your extensive correction of this manuscript. I really appreciate the work you did. I accept these changes and I am grateful for the clarification of some issues.

As I suggested previously, this article would have greater scientific value with more results from other organs collected from experimental animals, but I accept and understand your explanation concerning logistic and financial reasons. Hopefully, these analyses will be available and can be published in future.

However, there are still some necessary corrections to be done. 

Please add to you results (at least) small paragraph about the condition of experimental animals. You mention physical examination of pigs (line 222), but there are no results commenting this. You should inform if there were any problems with pigs’ health during the course of experiment, with the special emphasis on pigs’ status after the challenge (any clinical symptoms, local reactions at the site of vaccine administration, etc.).

Response: It is done. We thank the reviewer for this observation.

In the result section on line 410 we stated

“It is important to mention that no clinical symptoms associated with PRC were observed after inoculation and throughout the course of the experiment”. 

Also, discussion section we added in the line 792“It is important to mention that no side effects were observed at the injection site in the case of the injectable formulation or mucous membrane inflammation in the oro/nasal formula. This makes ChS-based microparticles a safe vehicle for animal vaccination”. 

Also, you used lungs homogenate as inoculum, so since these organs were collected from the diseased pigs, you should also expect presence of other respiratory tract pathogens (viral, bacterial) in them. Therefore, I suggest to correct the sentence starting in the line 244: “In week 7 after starting the experimental treatments (T1), the challenge was carried out by inoculation the four study groups (table 2) with lung homogenate as a source of M. hyopneumoniae, M. hyorhinis and PCV2.” Consequently, please mention in the Discussion that the inoculum has not been tested for the presence of other pathogens (or please correct me if I’m wrong), and, most probably, it contained a “cocktail” of microorganisms. All of these must have influenced the results.

 Response: It is done. We thank the reviewer for this observation.

In Mat y Met section on the line 242 we added “Additionally,  the inoculum macerate was checked by PCR negative for porcine reproductive and respiratory syndrome virus (PRRSV), swine influenza virus (SIV), Porcine Parvovirus and Actinobacillus pleuropneumoniae“.

Furthermore , in line 772 at Discussion section we stated “Although the inoculum used for the challenge (obtained from a lung macerate) was tested as negative for porcine reproductive and respiratory syndrome virus (PRRSV), swine influenza virus (SIV), Porcine Parvovirus and Actinobacillus pleuropneumoniae, it may have contained other not detected respiratory pathogens that may have interfered with the proper interpretation of the results

Comments on the Quality of English Language

  1. Please correct typo errors:
  • line 279: “lympnonode”

Response : It is done

  • line 401: “undercontrolled”

Responsw: It is done

  • line 403: “Threfore”

Response : It is done

  • line 411: please use “animals” instead of “animal”

Response: It is done

  • line 472: “titerss”

Response: It is done

  • line 781: “fr”
  1. line 428: please replace “lymph node” with “lymphoid”

Response: It is done

  1. line 485: please use “presenting” instead of “present”

Response: It is done

  1. Ple Response: It is done

ase correct fig. 2 and 3: you still have “titles” there…

  1. Please remove unnecessary commas in lines: 60, 589, 606.

Response: It is done

  1. Please check again the whole text for any other editioral mistakes.

Response: It is done

Reviewer 2 Report

Comments and Suggestions for Authors

The authors addressed the vast majority of the comments. However, as shown in Table 2, they administered a vaccine booster seven days after the experimental challenge. The reasoning for this decision should be explained in the manuscript.

Author Response

April  15th, 2024

Ms. Tessa Wang

Assistant Editor

MDPI Vaccines Editorial Office

Ref. Manuscript ID:  vaccines-2919578

Dear Editor,

Please find enclosed our responses to the reviewer’s comments for the manuscript “Titled: “Design of a New Vaccine Prototype Against Porcine Circovirus Type 2 (PCV2), M. hyopneumoniae and M. hyorhinis Based on Multiple Antigens Microencapsulation with Sulfated Chitosan”.
by: Darwuin Arrieta-Mendoza, Bruno Garces, Alejandro Hidalgo, Victor
Neira Ramirez, Galia Ramirez, Andrónico Neira-Carrillo, Sergio A Bucarey. The authors appreciate all comments provided by the referees, which have greatly helped us in reviewing our article and highlight the main aim of our research. Therefore, we feel that this new version of the manuscript is much clear and highlight the novelty of our scientific work.

All changes made in the text are in accordance with the reviewer's comments. Our point-by-point responses to each of the reviewer's comments are included below and are denoted in italics.

We believe our manuscript has been significantly improved by introducing the above-mentioned modification. We really appreciate your valuable opinion and guidance.

Sincerely yours,

Corresponding Authors

Sergio A. Bucarey

Response to Reviewers Comments 2

The authors addressed the vast majority of the comments. However, as shown in Table 2, they administered a vaccine booster seven days after the experimental challenge. The reasoning for this decision should be explained in the manuscript

 Response: We thank the reviewer for this observation.

Therefore , in Discussion section in the line 655 we added :

 “Another difference with the previous study [36], is that this prototype vaccine was not intended as a parenteral vaccine, emphasizing the fact that the vaccine has greater advantages when administered O/N due to its mucoadhesive properties of ChS. However, the low stability of antigens exposed to the harsh conditions in the gastrointestinal tract, together with the induction of mucosal tolerance, make difficult the induction of a reliable immune response through the oral/nasal delivery. Thus, two boosters were required in this study at 21 and 42 days after the 1st immunization, the second one 7 days after the experimental challenge